# Heated Behaviour in the Classroom for Children with FASD: The Relationship between Characteristics Associated with ADHD, ODD and ASD, Hot Executive Function and Classroom Based Reward Systems

**DOI:** 10.3390/children10040685

**Published:** 2023-04-04

**Authors:** Andrea Carrick, Colin J. Hamilton

**Affiliations:** Department of Psychology, Northumbria University, Newcastle upon Tyne NE1 8ST, UK

**Keywords:** FASD, children, ODD, ADHD, ALT, executive function, regulation, dimensional

## Abstract

Possession of characteristics related to Attention Deficit Hyperactive Disorder, Oppositional Defiance Disorder, and Autism Spectrum Disorder in children prenatally exposed to alcohol contributes to challenges within the diagnostic pathway for Foetal Alcohol Spectrum Disorder (FASD). The presentation of these characteristics, though problematic for the children affected, may not result in referral for diagnosis; focusing on diagnostic thresholds masks the dimensional nature of these characteristics. Children with traits which are undiagnosed may not receive effective support and are often identified as exhibiting challenging behaviour. In the UK, children with undiagnosed Special Educational Needs (SEN) are more likely to experience school exclusion. Common across each condition are challenges to executive function associated with emotional regulation (hot-executive function). This study explored the relationship between characteristics of Attention Deficit Hyperactive Disorder, Oppositional Defiance Disorder, Autistic-Like Traits, and hot executive functions on the helpfulness of reward-based interventions for children with suspected or diagnosed FASD. Data were collected online using caregiver referral questionnaire screeners for each measure (Child Autism Quotient Questionnaire, Vanderbilt ADHD Parental Rating Scale and The Childhood Executive Functioning Inventory) for children aged 6–12 years with suspected or diagnosed FASD (*n* = 121). Between-group comparisons showed no significant difference in the reporting of Attention Deficit Hyperactive Disorder characteristics, Oppositional Defiance Disorder characteristics, Autistic-Like Traits, and executive function, regardless of diagnostic state. Multiple regression analyses indicated that these personality characteristics and executive functions were associated with the perception of the reward system helpfulness. However, this pattern was qualified by both the type of hot executive function challenged (significant for Regulation not Inhibition) and whether the child had an FASD diagnosis. Thus, a dimensional approach may strengthen our understanding of the child’s classroom experience and help overcome barriers to effective intervention and support.

## 1. Introduction

Foetal alcohol spectrum disorder (FASD) is a neurodevelopmental disorder resulting from prenatal alcohol exposure, (PAE) [1]. As a teratogen, alcohol is an agent that crosses the placenta unfiltered to the foetus. The neuro-teratogenic effect of alcohol is multi-faceted; in terms of the amount and duration of exposure, and also the timing of exposure relative to the developmental stages of the foetus [2]. As a result, the damage to the foetus can be widespread, to encompass physical and neurodevelopmental birth defects that are difficult to detect and span a lifetime [3].

### 1.1. FASD and Problematic Diagnosis

Diagnostic pathways for FASD are inconsistent and patchy due to the historic lack of UK legislation. This is complicated further for clinicians by the lack of physical indication of FASD for the majority of those affected (present in less than 10%) and a reliance on confirmation of alcohol consumption during pregnancy, which is often lacking due to stigma or fear of ramifications. This can result in unpredictability in receiving a diagnosis, or the correct diagnosis [4,5,6]. A recent UK prevalence study indicated that between 1.8% and 3.8% of the population may be affected by FASD; up to 6% when taking into account suspected cases of FASD, with 66.1% males affected compared to 33.9% of females [3,7].

The possession of characteristics associated with Attention Deficit Hyperactive Disorder (ADHD), Oppositional Defiance Disorder (ODD), and Autism Spectrum Disorder (ASD) in children prenatally exposed to alcohol contributes to challenges within the diagnostic pathway for FASD. The high prevalence of FASD co-occurring with ADHD, ODD, and ASD can complicate the presentation of FASD, leading to missed or misdiagnosis [7]. Lack of diagnosis does not mean lack of impact.

Recent research indicates that pupils with neurodevelopmental difficulties are more likely to experience school exclusion [8]. The possession of ADHD, ODD, and ASD characteristics in children with FASD, though problematic for children affected, may not result in referral for diagnosis assessment. With waiting times for neurodevelopmental assessments in the UK, increasing the reliance on clinical diagnosis to receive support in the classroom may be having an effect on the child’s health, wellbeing, and learning [9].

### 1.2. Undiagnosed SEN and Exclusion

Unidentified Special Educational Needs (SEN) is a problem associated with school exclusion. With 55% of those permanently excluded, the identification of SEN only occurred during the exclusion period [10]. This means that the child’s support needs were initially missed or misunderstood, leading to exclusion. Children with identified SEN who have formal support are less likely to experience exclusion [11]. This highlights the importance of recognizing an unmet need in the presentation of pupils’ behaviour, which may be central to the identification and implementation of effective support strategies that improve education inclusion [12].

Due to lack of physical features associated with FASD, the first indication can be through the presentation of behaviour displayed by the child. These are behaviours which when misunderstood can become problematic within the classroom. Thus, many teachers may have taught students with FASD without knowing. The uneven developmental profile of FASD gives rise to the need for appropriate support to access Key Stages of the English National Curriculum [13]. Though IQ score may sit within normal levels, academic abilities are often well below. Difficulties with comprehension, social skills, and emotional maturity are often masked by physical maturity and advanced expressive language [13]. This can make behaviour misunderstood, leading to well-intended support strategies becoming ineffective, which may result in exclusion within or external to school.

Though there is an increase in the recognition of the conditions by clinicians, teachers, and caregivers [14], the reliance on a child coming into the contact with an educational practitioner who has specific knowledge of any of the conditions, or the impact of the overlap of them is compromised. This can lead to misinterpretation and misdiagnosis of the condition, resulting in inappropriate classroom interventions aimed at supporting pupils [12].

When school systems throw up difficulties for children and the child’s behaviour is not recognised by practitioners as potentially being characteristic of a clinical condition, or indeed the child lacks possession of a formal diagnosis of any SEN condition, the narrative of ‘problem’ quickly emerges [15]. Consequently, this leads to a higher risk of suspension and exclusion.

Those prenatally exposed to alcohol (PAE) can struggle to access diagnostic services for FASD [16], not least because of the invisible nature of a neurodevelopmental condition. Lack of diagnosis leads to lack of support, which increases the risk of health and socio-economic exclusion throughout the life span [4]. To address this, the NICE guidelines [17] recommended improvements to the diagnosis and support for children with Foetal Alcohol Spectrum Disorder (FASD), including a consideration within an educational setting.

### 1.3. A Dimensional Approach

Whilst understanding behaviour is important when supporting children in the classroom, the focus on diagnostic threshold to provide understanding can be problematic, as (1) lack of diagnosis does not mean lack of impact, and (2) long waiting lists for assessment mean a lack of interim support whilst waiting.

Although clinical assessments, including cognitive tests, provide robust evaluation of the child’s difficulties, caregiver reports are required in identifying thresholds for referral for clinical assessment. However, such measures can be seen as problematic. Social cognition of caregivers is required to interpret the behaviour and affective expressions of their children, raising concern for a ‘halo effect’ in reporting [18]; an unintentional bias stemming from their own social knowledge to shape their understanding of their child. However, the caregiver is often the first perspective in helping to understand the impact coming from undiagnosed conditions, and such assessments form part of the continuum of measurement used in diagnosis [19].

Conventionally, the assessment within neurodevelopmental conditions often leads to a focus on diagnostic thresholds for each independently co-occurring condition, missing the overlap across different conditions, along a spectrum of characteristics. The view through a diagnostic lens may complicate the assumption of effective interventions designed to support those affected [20]. When interventions are ineffective, the focus may shift to the problem lying with the behaviour of the individual rather than their support-need [12], leading to social, economic, and health exclusion.

The focus on diagnostic threshold assumes a homogenous approach to understanding the cognitive and behavioural characteristics of a heterogeneous population [20]. A trans-diagnostic approach, which aims to understand clusters of behavioural symptoms which may not map on to a diagnosis, shifts the focus from diagnostic-understanding to a need-based one [21]. Moving beyond the clinical silos has the potential to shape effective interventions to support individuals by meeting their need where there is a challenge to their daily functioning, rather than based on diagnostic labelling.

### 1.4. Co-Occurring Diagnoses and Executive Function

FASD, ADHD, ODD, and ASD are known to share similarities in patterns of challenge to executive function [6,21,22,23]. Cognitive processes in the form of executive function (EF) are essential to complete daily life tasks. Challenge to EF such as working memory and planning (cold EFs) and inhibiting and regulating behaviours (hot EFs), can lead to problematic interactions within education, the workplace, socially, and throughout the life course [24].

The effect of age and gender on each of these conditions when they co-occur is uncertain. The effects of ADHD may reduce with age [25]; however, deterioration in hot executive function performance for those with ADHD has been found to occur with aging and in females [26]. ASD characteristics may persist across the life span with an increased challenge to social communication [27]. Such a challenge is the same across genders; however, females show less repetitive and stereotyped behaviour [28]. The independent occurrence of ODD has been found to be invariant across age and gender [29]. The effect of gender and age when these conditions co-occur with FASD is unclear.

Conventionally, clinical EF research focuses on differences between clinical groups and typically developing groups; there is little insight into the impact of challenge to EF *within* undiagnosed groups. Lack of diagnosis for neurodevelopmental conditions can occur either because of (1) lack of referral for assessment (unassessed), (2) possession of characteristics may not reach a referral threshold (sub-referral), or (3) possession of characteristics when assessed by a clinician do not meet diagnostic criteria (sub-clinical). Given the prevalence of SEN diagnosed during school exclusion periods, it is important to understand the effect of challenges to EF along the continuum of characteristics associated with clinical conditions, regardless of diagnostic state. This is especially important when considering interventions aimed at shaping positive behaviour to promote classroom inclusion.

### 1.5. Rewards and Executive Function

Challenge to psychological processes associated with characteristics of ADHD, ODD, and ASD, regardless of diagnosis, may pose an elevated risk of anti-social responses to rewards and consequences. The implications of this have been found in the school-exclusion-to-prison-pipeline [8], which indicates that those with undiagnosed neurodevelopmental conditions, including FASD, are over-represented in both school exclusion and the youth justice system. The latter is a system which is built on reward and sanction as a way of regulating behaviour. Neither education, nor youth justice screen for neurodiversity, i.e., the characteristics of a condition without a diagnosis [12], leading to limited staff training. Knowledge gaps may contribute to a lack of support and identification of the ways challenge to psychological processes may impact on behaviour, in both these contexts.

Behaviour change mainly requires inhibiting and regulating undesired behaviour to shape desirable behaviour; inhibitory control and regulation are key executive functions [24]. If in a child there are challenges to these EFs, well intentioned classroom interventions aimed at managing self-regulation and educational inclusion can be problematic.

Shaping behaviour within school is often performed using reward-based incentive models which carry a social and moral common ground to encourage desired positive behaviour [30]. Reward and behavioural systems, intended as positive behavioural modelling, become problematic for those children who find it difficult to comprehend and maintain the expectations of emotional and behavioural regulation required within the classroom, leading to exclusions [15,31,32]. The personal meaning and emotion tied up in the motivation to achieve the reward evoke hot EF to navigate such challenges, processes such as inhibiting and regulating emotional responses. Challenges in hot executive function can lead to reward-based strategies being problematic in those with FASD, leading to high levels of frustration [33,34]; this is exacerbated when a co-occurrence of ODD, ADHD, and ASD characteristics are present.

For those with ODD, independent of ADHD, challenges in both hot and cold EFs have shown implications toward reward-related abnormalities in theories of anti-social behaviour development, related to increased risky decision-making and slower speed of inhibitory response [35]. There is a lack of knowledge on the effect of interventions targeting reward-related inhibitory control (associated with hot EF) for those with ADHD and ODD [36]. Reward-based decision making for those with ASD have been found to be problematic due to the exhibition of repetitive symptoms, particularly in relation to the need for sameness, and social impairment; the findings suggest an autonomic reactivity in response to reward and the anticipation of behavioural consequences [37]. Effective self-regulation strategies aimed at improving classroom exclusion require an understanding of the psychological capabilities of those they are aimed it [30]. Interpreting behaviours without a full understanding of the challenge to psychological processes underpinning them may lead to inappropriate interventions and misinterpretation of their effectiveness, ultimately leading to ineffective conclusions.

### 1.6. Aim and Rationale

Adopting a dimensional-transdiagnostic approach to understanding the impact of shared characteristics of multiple conditions along the continuum of diagnostic measurement, from sub-referral to clinical diagnosis, may help understand how daily functioning may be challenged, regardless of diagnostic status.

Children with undiagnosed FASD may not receive effective support and are often identified as exhibiting challenging behaviour. A global prevalence study indicated that those with FASD are represented 10–40 times higher than the general population in special education [1]. Given the prevalence of school exclusion in pupils with undiagnosed SEN, and the problematic FASD diagnostic pathways for children prenatally exposed to alcohol, investigating the co-occurring impact along the continuum of undiagnosed to diagnosed would be informative. This dimensional approach to understanding could explore the impact to the helpfulness of rewards for children in the classroom who possess elevated levels of ODD, ADHD, and ASD characteristics (subsequently in this paper labelled Autism-Like Traits, ALTs, when describing variance in the characteristics found across the general population). This dimensional approach moves the research away from a reliance upon diagnostic status to one which may be more productive in identifying effective strategies aimed at classroom inclusion.

As hot executive function is important to the effectiveness of reward-based behaviour-shaping interventions, then it is important to understand the effect when EF is challenged. Given the prevalence of overlap between ADHD, ODD, and ASD in children with FASD, exploration is required into the relationship between challenges to hot executive function and co-occurring ADHD, ODD, and ALT characteristics for those with FASD, regardless of diagnostic state of any of these conditions. Given that research has shown that increased practitioner awareness of these characteristics beyond any diagnostic label may lead to more effective classroom support for the child, this may lead to reduction in exclusion [38].

The evidence suggests that reward systems may be problematic for children with an FASD diagnosis. This study, employing caregiver ratings, aimed to examine the relationships between ADHD, ODD, and ALT characteristics and the child’s experience of reward systems. It further asks whether these relationships may be influenced by individual differences in emotional regulation and inhibition, and in turn, whether this mediation is affected by the FASD diagnostic status of the child.

The prediction for this study is that variability in hot executive function will mediate the observed relationship between characteristics of ADHD, ODD, or ALT and the experience of rewards. This mediation in turn may be affected by FASD diagnostic status.

## 2. Methods

### 2.1. Participants

To ascertain the sample size, an *a priori* power analysis was conducted using G*Power. Prior research [26] has typically found a large effect of ADHD and executive control, but with diagnostic levels of ADHD; this study was investigating children with lower levels of ADHD characteristics, so a medium effect size was adopted. This lead to a G*Power-calculated minimum of *n* = 53. This value was derived from a G*Power analysis, selecting regression analyses with 5 predictors, 3 of particular interest, and 2 co-variates.

Three hundred and thirty-six participants were initially recruited to take part in the online study using purposive sampling, based on being parents or carers (age 18 and above). The eligibility criteria for the study was parents and carers (caregivers) of children aged 6–12 years with suspected or diagnosed FASD. Recruitment was via National FASD and FASD Hub Scotland online support groups and associated networks. The online survey was made available to participants from April 2022 until September 2022. Only surveys where all questionnaires had >90% completion rates were eligible to be used within the study. In addition, an incomplete response to the age, gender, and FASD diagnosis status also led to exclusion. One hundred and twenty-one completed surveys were used to create the sample for the analysis, exceeding the G*Power recommendation. Chi-square comparison of the sample of caregivers who completed the survey, versus those who did not, indicated no difference in the gender proportions: *X^2^ = 3.36*, *p > 0.05.* However, the proportions with an FASD diagnosis did differ: *X^2^ = 6.403*, *p = 0.011*. This indicated that a greater proportion of caregivers whose child did not have an FASD diagnosis failed to complete the survey. The F test analysis comparing the age of the children whose caregiver completed the survey, versus those who did not complete the survey after recording their child’s age, did not differ significantly: *F (1, 56) = 1.354*, *p > 0.05*.

The sample included 67 children (24 girls, 1 non-binary, 42 boys) with diagnosed FASD (Mage = 9 years, SD = 1.8). A total of 54 children (27 girls, 1 non-binary, 26 boys) made up the suspected FASD group (Mage = 9 years 6 months, SD = 2.0). Chi-square analysis of the gender proportions in the two FASD diagnostic groups indicated that the proportions of boy and girls did not differ across the two groups, *X^2^ = 2.547*, *p > 0.05*.

In the FASD Diagnosed group, 47 of the children were reported as having a co-occurring diagnosis of ADHD, ASD, or ODD. In the FASD Suspected group, 20 of the children were identified as having a co-occurring diagnosis of ADHD, ASD, or ODD. Chi-square analysis revealed that the proportion of children with a co-occurring diagnosis did differ in the two FASD groups, *X^2^ = 13.267*, *p* < 0.001. Children in the FASD Diagnosed group were more likely to have co-occurring diagnoses.

### 2.2. Design

The study utilized a cross-sectional quantitative non-experimental design. An online questionnaire via the Qualtrics online platform (Qualtrics, Provo, UT, USA) was used to obtain the data. Data were collected through caregiver ratings of characteristics associated with ADHD, ODD, and ALTs, and ratings of the hot executive functions, regulation, and inhibition. Caregiver ratings of the helpfulness of rewards for their children were also collected. Diagnostic status related to FASD, ADHD, ODD, and ASD was collected. The online questionnaire presented measures of: EF using the Childhood Executive Functioning Inventory, (CHEXI) [39]; ALTs using the Autism Spectrum Quotient, (AQ-Child) [40]; and ADHD (all sub-types) and ODD using the NICHQ Vanderbilt Assessment Scales, (VADPRS) [19].

A number of analyses were performed to understand the interactions between the variables and how those relationships were associated with the helpfulness of rewards (see Section 3.1). Due to the prevalence of co-occurring ADHD, ODD, and ASD, the analysis strategy was to use a multi-variate approach to understand how observed variables (executive function, FASD diagnosis, age, and gender) may confound any meaningful interpretation of the relationship between characteristics of each condition and the helpfulness of rewards.

For the main moderated mediation analysis, the predictor variables were scores from the ADHD, ODD, or ALT measures. The outcome variable was ratings of the helpfulness of reward systems in schools. Given that the prior literature indicated a relationship between ADHD, ODD, ALT, and challenge to executive function, the score of executive functions (regulation or inhibition) was used as a mediator to understand its indirect effect in the relationship between the characteristics and rewards. Though the literature indicated a relationship between traits and executive function, it was possible that having an FASD diagnosis may account for any variance in any indirect effect found, and so the observed variable FASD Diagnosis (suspected or diagnosed) was used as a moderator on the indirect effect. Covariates were age and gender within all the analyses, due to prior research indicating their association with ALT and ADHD characteristics.

### 2.3. Materials

#### 2.3.1. Online Survey

The survey content began with a Participant Information form which obtained informed consent prior to undertaking the study. Participants were informed of their rights of withdrawal, anonymity, and data storage security. A Participant Debrief sheet, including how to access further support on FASD, ASD, ODD, and ADHD was made available at the end of the survey. Caregivers answered demographic questions related to their child, which included age, gender, school experience, and diagnostic status. Questions were positively worded and positively scored to reduce the risk of bias in reporting from a potential negative halo effect.

Three questionnaires were employed to measure caregiver reporting of variability in ALTs, (AQ-Child) [40], ODD and ADHD characteristics [19], and EF characteristics, (CHEXI) [39].

#### 2.3.2. Questionnaires

##### Childhood Executive Functioning Inventory (CHEXI)

The CHEXI [39] is a rating instrument used for measuring parental and teacher reporting of executive functioning in children aged 4–12, using a 5-point Likert scale with good psychometric properties [41]. It includes four different subscales of inhibition and regulation (Hot executive function) and working memory and planning (Cold executive function). The higher the score, the higher the indication of challenge to executive function. Given the overlap between ADHD and FASD, and CHEXI’s discriminate ability [41], it was used in this study to measure the four subscales of executive function. For this paper, only inhibition and regulation will be focused upon. The original reported reliability was found to be adequate (α = 0.89). The current study reliability for the regulation and inhibition components was α = 0.62, and α = 0.73, respectively.

##### Child Autism Spectrum Quotient (AQ Child)

The AQ Child is a parent-report questionnaire conventionally used to measure the expression of ALTs in children aged 4–11 years [40]. Higher scores indicate a greater possession of characteristics similar to those present in ASD. The 4-point Likert scale has been shown to have good test-retest reliability and high internal consistency and was used in this study to measure ALT characteristics. The original reported reliability was high (α = 0.97); the current study reliability for the AQ was α = 0.91.

##### NICHQ Vanderbilt Assessment Scales (VADPRS)

The Vanderbilt ADHD diagnostic parent rating scale is conventionally used as a screening tool to help in the diagnostic process of ADHD (both inattentive, hyperactive, and combined subtypes), in children aged 6–12 years [19]. It has 55 questions in total, and includes questions related to co-occurring characteristics of ODD, Conduct Disorder, Anxiety, and Depression. Higher scores indicate a greater possession of the characteristics. The 4-point Likert scale has high reliability and clinical utility [42]. This study split the subset of questions into ADHD and ODD in order to obtain separate measures for each variable. The original reported reliability for the scales was good (all α > 0.90), and the current study reliability for the Inattention and Hyperactivity components was α = 0.90, and α = 0.85, respectively.

##### Experience of Rewards

The Qualtrics survey asked 4 questions, using a 4-point Likert scale to measure parents’ observations of how helpful rewards were for their child: “never”, “occasionally”, “often”, “very often”. Three of the questions asked about helpfulness of rewards in different contexts: “instant” i.e., sticker for good work there and then; “time-lapse” i.e., certificate at end of week; and “recognition in comparison to others” i.e., certificate for highest attendance. The fourth question asked about the helpfulness of rewards overall. Cronbach’s alpha for the measure of the helpfulness of rewards was found to be marginally acceptable (α = 0.68).

### 2.4. General Procedure

Ethical approval for the study was granted by the Department of Psychology Postgraduate Ethics committee at Northumbria University (Reference number-46032). Links to the survey were sent to the support groups and posted online. A copy of the information sheet and informed consent was sought at the start of the survey. Participants were then asked to complete the demographic questions, general questions, and the questions from each of the behavioural checklists. Questionnaire items were presented individually onscreen, thus not only ensuring anonymity but also reducing any acquiescent response biases. After completing the survey, participants were debriefed and provided with the opportunity to withdraw their data, if they so wished. The survey took approximately 25 min to complete. Where participants emailed expressing interest in the results, they have been advised that a user-friendly version will be made available. Follow-up analysis is intended to explore with the participants the direction of future research.

## 3. Results

### 3.1. Data Treatment

Analysis on the total sample was performed to identify incomplete surveys. Any survey which contained >10% of unanswered questions from a questionnaire was removed from the data collection. In addition, surveys which omitted responses to either age, gender, or FASD diagnosis status were also removed. In the final sample of respondents (*n* = 121) missing data from each measure of ADHD, ODD, ALT, and EF were replaced with average scores.

The data were initially reported in terms of descriptive statistics of two groups, FASD Diagnosed and FASD Suspected, in measures of ADHD, ODD, ALT, EF and helpfulness of reward systems. Subsequently, one-sample *t*-tests on the measures were carried out to determine whether the measures displayed relatively high scoring in relation to each of their scale mid-points. This was followed by a series of ANCOVAs with age and gender as the co-variates, with Group as the main factor and the ADHD, ODD, ALT, hot EFs and helpfulness of reward systems measures. To explore the simpler relationships between the measures, partial correlations between traits and reward system (controlling for age and gender) were carried out. To understand the relationship between ADHD, ODD, ALT, and executive function, mediation analyses using Hayes PROCESS Model 4 [43] was considered and discounted, given that this was a non-experimental study which was considering the relationship between the observed variables rather than any perceived causal effect. PROCESS Model 14 was instead employed, allowing the fourth observed variable (FASD Diagnosis) to be added into the analysis. This moderated mediation analysis considered how EF and FASD diagnosis qualified the relationship between traits and the helpfulness of rewards, whilst controlling for the effect of age and gender. Regulation and Inhibition were individually considered as mediators of these relationships. The final step of these analyses included FASD diagnosis status, confirmed vs suspected, as a putative moderator of the mediation.

### 3.2. Findings

#### 3.2.1. Descriptive Statistics and One-Sample *t*-Tests

Table 1 below indicates the descriptive statistics for the Suspected and Diagnosed groups across all of the measures. For the personality trait and EF measures, the means appear relatively high for both groups; in order to inform this observation, a series of one-sample *t*-tests was carried out for each of the measures, comparing their distributions to the mid-point of each scale.

The findings revealed that for both groups, in all of the trait and EF analyses, the distribution of scores were significantly higher than each of their respective scale mid-points (all *ps* < 0.05). This indicates that for both groups there were challenges across all of the trait characteristics and in both hot executive functions (see Appendix A, labelled Full Data Analyses, for full details of these analyses).

#### 3.2.2. FASD Diagnosed and FASD Suspected Group ANCOVAS

To directly compare the possession of the traits, hot executive functions, and reward system experience between the two groups, a series of ANCOVAs was carried out with Group as the factor. In the comparison of ODD trait possession, the Group factor was not significant: F (1, 117) = 1.87, *p* > 0.05. For the Child AQ measure, the Group factor was not significant: F (1, 117) = 2.173, *p* > 0.05. For the VADPRS–ADHD Combined measure, the Group factor was not significant: F (1, 117) = 1.778, *p* > 0.05. For the VADPRS–ADHD Inattention measure, the Group factor was not significant: F (1, 117) = 3.742, *p* > 0.05. For the VADPRS–ADHD Hyperactivity measure, the Group factor was not significant: F (1, 117) = 0.435, *p* > 0.05. For the CHEXI Regulation measure, the Group factor was not significant: F (1, 115) = 0.375, *p* > 0.05. For the CHEXI Inhibition measure, the Group factor was not significant: F (1, 117) = 1.254, *p* > 0.05. Finally for the reward system ratings, the Group factor was not significant: F (1, 113) = 0.661, *p* > 0.05.

Thus, across all these analyses, there were no significant differences between the two groups in the level of trait possession, hot executive function, and the experience of reward systems.

#### 3.2.3. Partial Correlations

The next analyses investigated the partial correlations between the trait measures, hot executive measures, and the reward system helpfulness. The partial correlations are shown below in Table 2. Many of the variables in both the FASD Diagnosed and FASD Suspected groups show significant partial correlations with one another. However, one noticeable contrast is in the relationships with the reward system ratings. In the FASD Diagnosed group (above the diagonal), most of the trait and EF measures are significant predictors of the reward system rating scores; in the case of the FASD Suspected group, all these correlations are non-significant.

#### 3.2.4. Moderated Mediation Analyses

The final analysis looked at the interaction of the relationships between all observed variables; the traits, ADHD-Combined, ADHD-Inattention, ADHD-Hyperactivity, AQ, ODD; the hot executive functions, regulation and inhibition, and diagnostic status, FASD Diagnosed, and FASD Suspected (see Figure 1), through a series of moderated mediation analyses. The effect from age and gender was controlled for in all analyses.

The explicit rationale for the analysis is shown immediately below for ADHD-Combined, including tables of the model results where regulation is a mediator (Table 3 and Table 4). This is followed by a table summarising all the models where a significant moderated mediation occurred (Table 5). The bootstrapping value was set as 5000 (i.e., estimation of conditional indirect effect based on 5000 bootstrap samples) to measure the standard error.

In the model for ADHD-Combined, there was no direct relationship between ADHD-Combined and rewards (c’ path). The *a path* showed that ADHD-Combined had a significant effect on Regulation, and that neither age nor gender were significant in this relationship (all *ps* > 0.2358). The *b path* showed a variety of significant and non-significant results (see Table 3). Here, the relationship between executive function (regulation) and rewards (*b1*) was not significant, the relationship between FASD diagnostic state and rewards was significant (*b2*), and the interaction between Regulation and FASD diagnostic state had a significant effect on rewards (*b3*) (unadjusted estimate *R*_2_ = 0.045 *p* = 0.0197). There was no effect from age and gender in this model (all *ps* > 0.4311).

The analyses of the index of moderated mediation (*ab3*) found a significant indirect effect on the relationship between ADHD-Combined and rewards, which was conditional on the interaction between regulation and FASD Diagnostic state (coefficient = 0.449, SE = 0.218 CI 95% [0.095, 0.949]). This indicated that moderated mediation was occurring, and that from the observed variables the indirect effect of regulation was conditional on the FASD diagnostic state. The probe into the conditional indirect effect (*ab1 + ab3W*) showed that the conditional effect was significant in the FASD Diagnosed group (coefficient = 0.510, SE = 0.167 CI 95% [0.226, 0.874]) but not in the FASD Suspected group (see Table 4).

The model indicates that when analysing the moderating effect of FASD diagnosis on the indirect effect of regulation in the relationship between traits and rewards, the interaction between regulation and FASD diagnosis changes the significance of the statistical effect. It suggests that an increase in regulation score by one unit, related to an increase in the ADHD-Combined score (*a path*), will increase the score on the helpfulness of rewards by 0.499 (*ab3*), across the FASD diagnostic state. The results showed that the conditional indirect effect (moderated mediation) is significant in the FASD Diagnosed group not for the FASD Suspected group, increasing the effect on the score by 0.510. This indicates that the effect of ADHD-Combined on the helpfulness of rewards may is come indirectly via the interaction between FASD diagnostic state and regulation.

For all the models there was no direct relationship between the trait and the reward system, whether regulation or inhibition was the mediator in the model. The results from each moderated mediation analysis showed that there was no significant direct path X-Y, nor in the *b1* path M-Y, and therefore indicated that the effect on the helpfulness of rewards was not coming from either the characteristics of ADHD, ODD, ALT, or each executive function, independently. Rather, the indirect effect (*b2* path) was conditional on FASD diagnosis (*b3*). These results indicate a conditional indirect effect related to a diagnosis of FASD, rather than caused by it; for this group, rewards are more unhelpful when they have challenges to regulation which are related to challenges associated with characteristics of ADHD, ODD, or ASD.

The consistent result in each model was the interaction between regulation and FASD status having a significant indirect effect on the relationship between trait and reward, and the interaction between inhibition and FASD status being non-significant. The lack of moderated mediation when inhibition was in each model indicates that regulation, not inhibition, may be influencing the effect. The results shown in Table 5 show FASD Diagnosis moderating the mediation effect of Regulation in the relationship between each of the traits (X) and rewards (Y). In all models, Table 5 shows the conditional indirect effect of the moderated mediation as significant for the FASD Diagnosed group. Refer to Appendix A for significant results for all models.

##### Summary of Findings

The findings suggest that the possession of ALT, ODD, and ADHD characteristics were not significantly different in their intensity across the FASD Suspected and FASD Diagnosed groups. Critically, in the perceptions of the relationships between these traits and the child’s experience of rewards, the role of Regulation differed between the groups, only being significant for the FASD Diagnosed group.

## 4. Discussion

The aim of this non-experimental study was to understand how the relationship between hot executive function (regulation and inhibition) and characteristics of ADHD, ODD, and ALT impacts upon the helpfulness of reward-based behaviour-shaping interventions for children with FASD. Using caregiver reporting questionnaire measures, we assessed the variability in ADHD, ODD, and ALT characteristics; hot executive function; and the experience of rewards for children with FASD, both suspected and diagnosed. The study addressed the research question of how hot executive function mediates the relationship between characteristics of ADHD, ODD, and ALT and the experience of reward-based behavioural systems in school for children with FASD, and the extent to which this mediation may be dependent on the child having an FASD diagnosis.

The caregiver ratings across each of the ALT, ODD, and ADHD measures indicated a significantly elevated distribution of scores. This was also the case for the hot EF measures. Given that in all the trait and EF scales, higher scores indicate challenges or difficulties, then this sample of children is evidencing high levels of behaviour and emotional challenges.

A group comparison, contrasting the FASD Diagnosed and FASD Suspected groups, found no difference in the ratings of ADHD, ODD, ALT, EF, and reward helpfulness. This lack of difference in the possession of traits occurred despite the two groups significantly differing in the frequency of their co-occurring diagnoses. This finding, regardless of FASD diagnostic status (suspected and diagnosed), resonates with the suggestion that those with suspected FASD experience similar challenges to those with a diagnosis [3,7]. Importantly, the possession of these characteristics increases the risks of mental health [44,45,46] and cognitive challenges [47]. These children, *regardless* of their diagnostic status, are likely to encounter challenges within and outside of the classroom environment.

Multiple regression, using moderated mediation analyses (Hayes PROCESS Model 14) [43], indicated that the variability in levels of ADHD, ODD, and ALT characteristics was associated with variability in the caregiver perception of the helpfulness of rewards. However, from the observed findings, this relationship pattern is associated with both the EF characteristics and the whether the child had an FASD diagnosis. The consistent pattern of results found that, when measured by the caregivers of those with diagnosed FASD, Regulation was a significant mediator in the relationship between each of the traits and the helpfulness of rewards. For those whose children had been diagnosed with FASD, as the score on challenge to regulation increased, the perception of rewards being more unhelpful increased. The non-significant findings for inhibition as a mediator indicated that regulation, not inhibition, was the important variable in the indirect effect of executive function and FASD diagnostic state. The main findings were in line with the global prediction that challenges to hot executive function mediates the observed relationship between subclinical characteristics of either ADHD, ODD, or ALT, and the experience of rewards. This is consistent with the suggestions that challenges to hot executive function may account, in part, for the similarities in the presentation of these conditions [48].

The significant results from the moderated mediation analyses tell us a story of how the interaction of these measured variables is important for the helpfulness of rewards for children with FASD. However, as heeded by Hayes [43], they are not indicative of a causal relationship. As with any non-experimental study, causal inference can lead to an incorrect interpretation of the results. Rather, the conditional processing model allowed the consideration of how the variables may relate to one another, conditional on a fourth variable. Consequently, there must be an understanding that unobserved variables could putatively lead to a misinterpretation of the findings.

These findings support the argument that a dimensional approach, examining variability in ODD, ADHD, and ALT in the absence of a diagnosis, may strengthen our understanding of the child’s classroom experience.

Caregivers reporting their children as high along the continuum of measurement for regulation and inhibition, regardless of diagnostic state for any condition, supports the call to move away from the reliance on diagnostic state to understand the challenge to daily function faced by children who possess characteristics associated with diverse neurodevelopmental conditions [12,20].

The results from the caregiver screening questionnaires provide a good indication of the variability across the pre-diagnosed end of the dimension, highlighting the value in the caregiver’s assessment of challenges faced by their child. The higher frequency of diagnosed co-occurring conditions in the FASD Diagnosed group provides a greater confidence in the variability of characteristics at the diagnosed end of the dimension and could account for the significant findings for this group in the moderated mediation analyses.

The moderated mediation analysis supports the importance of a dimensional-transdiagnostic approach in understanding how challenges in daily functioning may shift based on context. This analysis highlights that, though both regulation and inhibition are challenged, in the context of rewards it is probably the effect coming from emotional and behavioural regulation, not inhibition, which leads to rewards being unhelpful. In this context, challenge to regulation was a consistent pattern regardless of diagnostic state. This is an important factor for when practitioners are selecting interventions to shape behaviour within the classroom.

Challenge to psychological processes in the form of hot executive function may increase the risk of anti-social responses to reward and consequence. These findings may be of interest in the context of the youth justice system, where young people with ADHD, ASD, and FASD are over-represented, many of whom may have no diagnosis [8]. The heavy reliance on reward and consequence to shape behaviour within the system, may be ineffective due to the child’s inability to comprehend moral and social norms tied to the rules [30], increasing levels of frustration, resulting in anti-social responses. Lack of knowledge regarding the challenge to psychological processes may lead to punitive consequences and further criminalization [12].

It should be understood that challenges in emotional and behavioural regulation may be important when considering reward systems for children with FASD. Focusing on the challenge to regulation processes, rather than the diagnosis status, may help educational practitioners when selecting interventions to encourage positive classroom behaviour [8,9,17].

### 4.1. Limitations

Caregiver assessment without other measures such as cognitive tasks limit the findings of this study to the caregiver’s perception and social cognition. However, concerns regarding bias known to be associated with caregiver reporting of challenging behaviour of their children (e.g., the halo effect) [49] were addressed in the findings. A non-significant group difference in the reporting of the helpfulness of rewards indicated consistency in how caregivers viewed the child’s experience of rewards, despite the FASD diagnostic status. The pattern of results suggested that there was variability in the nature of the relationships, e.g., with only regulation acting as a significant mediator, and not inhibition. If a halo effect was the major driver in the data one may have expected a greater level of consistency in the pattern of results.

As indicated previously, latent variables uncontrolled for in this non-experimental study could account for some of the variance observed and therefore the results from this study should be inferred as indicative rather than definitive.

The data does not indicate the child and caregivers experiences of the diagnostic pathway, so we cannot assess whether this influenced their ability to interpret their child’s behaviours. The national inconsistency across the UK in diagnostic pathways and subsequent support received within and beyond the assessment process [50] makes it difficult to interpret the likelihood of the carer’s knowledge influencing their scores.

### 4.2. Future Research

Given that behaviour-shaping interventions are sensitive to the environment in which they are performed [31], the information from this study is limited to assessing caregiver perception. Understanding the experience of interventions is important in exploring the ways in which practitioners interpret the behaviours of children who have no formal diagnosis. Thus, exploring the educational practitioners’ perception of these characteristics in children could help in understanding the journey from behaviour being displayed to interventions being implemented, and the outcomes associated with the effectiveness of intervention. Qualitative research to understand the child’s, classroom practitioner’s, and the caregiver’s experience of reward systems may help to improve our overall understanding of this process.

Though the caregiver measures used in this study have been found to have good reliability, further study would benefit from understanding the psychological mechanisms which are associated with these co-occurring characteristics. Performing cognitive tasks (in addition to rating scales) to compare the findings in relation to reporting by caregivers, non-clinical, and clinical professionals could provide a more robust understanding of the psychological mechanisms underpinning the challenges experienced [51].

Welcome advances in theoretical models that aim to improve support for the most vulnerable school children, such as trauma-informed approaches [52], may go some way to addressing the support needs of children with FASD. These approaches acknowledge that psychological processes such as regulation and inhibition can be challenged because of pre-and post-natal exposure to trauma, providing a broader approach that goes beyond a specific diagnostic understanding. Currently, the number of empirical findings on the efficacy of trauma-informed models for children with FASD is limited.

Educational approaches specific to children with FASD are emerging in the UK, with information and training for educators which have been developed in conjunction with those affected by FASD; however, there have been limited empirical studies involving practices towards FASD in educational settings [53]. As a result, there has been little effective intervention identified for children with FASD [54]; further evidence-based exploration is required in this field. Co-production models of intervention which harness stakeholder involvement from students and educators have been found to be effective in improving issues such as mental health, psychological well-being, and school inclusion. Psychological theory to underpin the intervention [55] and peer-involvement [56] has led to successful interventions which benefit cohorts of students in the classroom regardless of diagnostic condition. Taking a heath-based approach to identifying barriers to interventions in educational settings, and ways to overcome them, such as the Behaviour Change Wheel taxonomy [30] could be beneficial.

Incorporating these approaches, a mixed-method study could help to explore how considerations of psychological mechanisms related to characteristics associated with neurodevelopmental conditions such as ADHD, ODD, and ASD can present in the classroom. Providing psychological underpinning may improve the understanding of behaviour for those with undiagnosed SEN, reducing the reliance on assumptions of causes of behaviour [15]. It may help to identify effective support for children who are waiting for an assessment for diagnosis, and in turn improve targeted interventions aimed at improving the educational experience of children with FASD.

Working collaboratively with children, their families, schools, and FASD organisations would provide the potential for co-designed interventions which may empower the FASD community.

## 5. Conclusions

This study has found executive function, in the form of emotional and behavioural regulation, to be an important variable to be considered when using reward-based behaviour-shaping strategies. It has reflected upon how a dimensional approach to understanding the characteristics of co-occurring neurodevelopmental conditions has the potential to enhance understanding and support for those with FASD, an area of improvement as recommended in the NICE guidelines [17]. It highlights the importance of exploring the effects along a continuum of measurement, as a way of attempting to understand unexpected behaviours within the classroom. In moving away from a focus on clinical thresholds, this approach in future research could improve understanding of the psychological mechanisms that underpin effective behaviour-shaping interventions for children with undiagnosed SEN [57]. This, along with future research to understand the experiences of FASD within the classroom, has the potential to influence how children can be supported whilst awaiting assessment for a SEN diagnosis. This could help to address the issue of reducing school exclusion; known to be associated with long-term inequalities in economic, health and social outcomes for children with FASD.

## Figures and Tables

**Figure 1 children-10-00685-f001:**
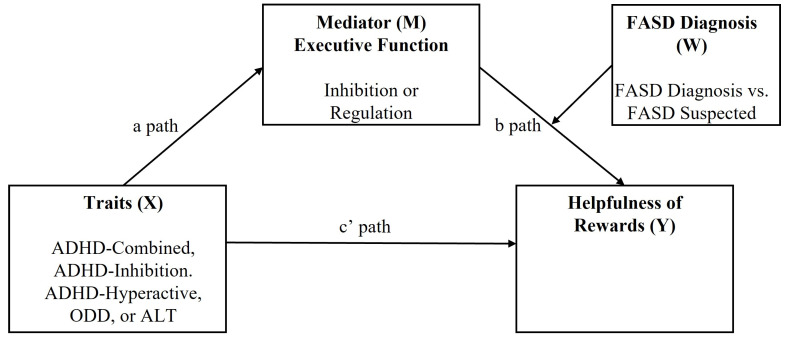
Hayes Model 14: Moderated Mediation Model.

**Table 1 children-10-00685-t001:** Descriptive statistics showing the ratings of the Suspected FASD and Diagnosed FASD groups across the various measures.

	Suspected FASD				Diagnosed FASD			
Variable	N	M (*SD*)	Min	Max	N	M (*SD*)	Min	Max
Reward	51	2.3 (0.6)	1	3.25	66	2.3 (0.5)	1	3.25
VAS ODD	54	2.2 (0.6)	0.75	3.00	67	2.1 (0.6)	0.57	3.00
Child AQ	54	1.7 (0.4)	0.64	2.58	67	1.9 (0.4)	1.10	2.76
VAS ADHD Inhibition	54	2.4 (0.6)	0.89	3.00	67	2.6 (0.4)	1.44	3.00
VAS ADHD Hyperactivity	54	2.2 (0.7)	0.56	3.11	67	2.3 (0.6)	0.56	3.11
VAS ADHD Combined	54	2.3 (0.6)	1.00	3.06	67	2.5 (0.5)	1.11	3.06
CHEXI Regulation	54	4.5 (0.5)	3.00	5.00	65	4.7 (0.4)	3.40	5.00
CHEXI Inhibition	54	4.3 (0.6)	2.83	5.00	67	4.5 (0.6)	2.50	5.00

*Note*. Reward: Reward System; VAS–ODD: Vanderbilt Assessment Scales–Oppositional-Defiant Disorder; Child AQ: Child Autism Spectrum Quotient (AQ); VAS–ADHD Inhibition: Vanderbilt Assessment Scales–ADHD Inhibition; VAS–ADHD Hyperactivity: Vanderbilt Assessment Scales–ADHD Hyperactivity; VAS–ADHD Combined: Vanderbilt Assessment Scales–ADHD Combined; CHEXI Regulation: Childhood Executive Functioning Inventory–Regulation; CHEXI Inhibition: Childhood Executive Functioning Inventory–Inhibition.

**Table 2 children-10-00685-t002:** Partial correlations controlling for age and gender between the traits and the reward system helpfulness.

		Reward	CHEXI Regulation	CHEXI Inhibition	VAS ODD	Child AQ	VAS ADHD Combined	VAS ADHD Inattention	VAS ADHD Hyperactivity
Reward	*r*		0.444	0.276	0.276	0.436	0.241	0.322	0.153
	*p*		<0.001	0.03	0.03	<0.001	0.06	0.011	0.235
CHEXI Regulation	*r*	0.004		0.343	0.299	0.47	0.49	0.618	0.338
	*p*	0.978		0.006	0.018	<0.001	<0.001	<0.001	0.007
CHEXI Inhibition	*r*	0.063	0.617		0.452	0.325	0.757	0.665	0.729
	*p*	0.668	<0.001		<0.001	0.01	<0.001	<0.001	<0.001
VAS ODD	*r*	0.041	0.512	0.406		0.316	0.574	0.504	0.551
	*p*	0.778	<0.001	0.004		0.012	<0.001	<0.001	<0.001
Child AQ	*r*	−0.073	0.205	0.268	0.205		0.341	0.387	0.266
	*p*	0.616	0.157	0.063	0.157		0.007	0.002	0.037
VAS ADHD Combined	*r*	−0.089	0.545	0.507	0.44	0.097		0.897	0.949
	*p*	0.541	<0.001	<0.001	0.002	0.505		<0.001	<0.001
VAS ADHD Inattention	*r*	−0.009	0.523	0.426	0.395	0.099	0.879		0.712
	*p*	0.950	<0.001	0.002	0.005	0.498	<0.001		<0.001
VAS ADHD Hyperactivity	*r*	−0.139	0.467	0.483	0.398	0.079	0.922	0.625	
	*p*	0.341	<0.001	<0.001	0.005	0.590	<0.001	<0.001	

*Note*. Partial correlations above the diagonal refer to the FASD Group, below the diagonal to the FASD Suspected group. Reward: Reward System; VAS–ODD: Vanderbilt Assessment Scales–Oppositional-Defiant Disorder; Child AQ: Child Autism Spectrum Quotient (AQ); VAS–ADHD Inhibition: Vanderbilt Assessment Scales–ADHD Inhibition; VAS–ADHD Hyperactivity: Vanderbilt Assessment Scales–ADHD Hyperactivity; VAS–ADHD Combined: Vanderbilt Assessment Scales–ADHD Combined; CHEXI Regulation: Childhood Executive Functioning Inventory–Regulation; CHEXI Inhibition: Childhood Executive Functioning Inventory–Inhibition.

**Table 3 children-10-00685-t003:** The model coefficients and model summary information for the conditional process (Model 14) in the relationship between ADHD-Combined and the helpfulnes of rewards, where FASD diagnostic status moderated the mediation effect of Regulation.

		Consequent						
		M (Regulation)				Y (Rewards)		
Antecendent		Coeff.	SE	*p*		Coeff.	SE	*p*
X (ADHD-C)	a	0.198	0.034	<0.001	c‘	−0.120	0.224	0.5915
M (Regulation)		--	--	--	b1	0.312	0.746	0.6769
W (FASD status)		--	--	--	b2	−10.933	4.465	0.0160
M X W		--	--	--	b3	2.268	0.958	0.0197
Constant	im	3.416	0.237	<0.001	iy	7.769	3.166	0.0157
		R^2^ = 0.258				R^2^ = 0.133		
		F(1, 114) = 12.746, *p* < 0.001				F(1, 114) = 2.737, *p* = 0.0164		

*Note*. ADHD-C: Vanderbilt Assessment Scale ADHD Combined.

**Table 4 children-10-00685-t004:** Showing the conditional indirect effects of regulation and FASD diagnostic state on the relationship between ADHD-C and rewards.

FASD Status (W)	Effect of X on M a	Conditional Effect of M on Y θM→Y = b1 + b3W	Conditional Indirect Effect of X on Y aθM→Y = a(b1 + b3W)
Suspected	0.198 *	0.312	0.061
Diagnosed	0.198 *	2.580 *	0.510 *

* This indicates a significant result.

**Table 5 children-10-00685-t005:** Moderated mediation results for the moderated mediation models with Regulation as the mediator.

		(M) Regulation (W) FASD Diagnosis				
					95% CI	
Traits (X)		Parameter	Coeff.	SE	LCI	UCL
VAS ADHD Combined	Moderated Mediation	(ab3)	0.449	0.218	0.095	0.949
	Conditional Indirect effect	(ab1 + ab3W)	0.510*	0.167	0.226	0.874
VAS ADHD Inattention	Moderated Mediation	(ab3)	1.094	0.504	0.255	2.264
	Conditional Indirect effect	(ab1 + ab3W)	1.132 *	0.370	0.452	1.892
VAS ADHD Hyperactive	Moderated Mediation	(ab3)	0.513	0.283	0.095	1.196
	Conditional Indirect effect	(ab1 + ab3W)	0.594 *	0.283	0.095	1.196
VAS ODD	Moderated Mediation	(ab3)	0.656	0.294	0.150	1.303
	Conditional Indirect effect	(ab1 + ab3W)	0.616 *	0.234	0.225	1.133
Child AQ	Moderated Mediation	(ab3)	0.762	0.448	0.097	1.810
	Conditional Indirect effect	(ab1 + ab3W)	0.793 *	0.331	0.252	1.541

* Significant result for FASD Diagnosed. *Note*. Child AQ: Child Autism Spectrum Quotient (AQ); VAS–ADHD Inhibition: Vanderbilt Assessment Scales–ADHD Inhibition; VAS–ADHD Hyperactive: Vanderbilt Assessment Scales–ADHD Hyperactivity; VAS–ADHD Combined: Vanderbilt Assessment Scales–ADHD Combined.

## Data Availability

Data available at OSF, Hamburg site DOI 10.17605/OSF.IO/YNHU4, accessed on 28 February 2023.

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
