# Peer review of "Heated Behaviour in the Classroom for Children with FASD: The Relationship between Characteristics Associated with ADHD, ODD and ASD, Hot Executive Function and Classroom Based Reward Systems"

_children, 2023, doi:10.3390/children10040685_

Round 1

Reviewer 1 Report

This paper presents an empirical study on heated behavior in the classroom for children with Fetal Alcohol Spectrum Disorder.

The research topic addressed is of high relevance for both research and practice.

The manuscript is well written and would nicely fit in the Special Issue “Advances in Fetal Alcohol Spectrum Disorders (FASD)”.

Important strengths are the different disorders considered and the practical relevance.

Given the null effects at some places: has an a priori power analysis been conducted?

Age may have influenced the pattern of results. Effects may be more pronounced in certain age subgroups.

Potential differences between boys and girls could be discussed in more detail.

The practical relevance in the school context, but also different environments could be illustrated in more detail.

The potential for novel interventions could be highlighted more strongly with more concrete examples.

Please consider elaborating in more depth on how current conceptual models are advanced. This could help to frame future research development.

Title: There are numerous abbreviations in the abstract. Please consider avoiding abbreviations for more clarity.

Author Response

Reviewer 1

Thank you very much for your comments, we are grateful for your suggestions on how to improve the quality of the manuscript and detail below our responses.

1)Given the null effects at some places: has an a priori power analysis been conducted?

An a priori power analysis was conducted. This has been referred to within the Method section. We have expanded the explanation to include:

“To ascertain the sample size an a priori power analysis was conducted using G*Power. Prior research e.g., Skogli et al., 2017, has typically found a large effect of ADHD and executive control, but with diagnostic levels of ADHD, this study was investigating children with lower levels of ADHD characteristics so a medium effect size was adopted. Leading to a g*Power calculated minimum n = 53. This value was derived from a G*Power analysis, selecting regression analyses with 5 predictors, 3 of particular interest, and 2 co-variates.”

2)Age may have influenced the pattern of results. Effects may be more pronounced in certain age subgroups.

We agree. Age was controlled for and mentioned in the design section and data treatment section. We have added a paragraph in the introduction which refers to the effect of age and gender, please see point 3.

3)Potential differences between boys and girls could be discussed in more detail.

We have added a paragraph to the introduction which comments on the effect of age and gender within this population (see below). However, gender difference was not the focus of this present study. We found that there is no reliable gender difference across the two FASD diagnostic groups. However, given a priori considerations, we controlled for the effect pf gender in all of the analyses. This is now made clearer in the Method and Results sections.

“The effect of age and gender on each of these conditions when they co-occur is uncertain. The effects of ADHD may reduce with age [25], however deterioration in hot executive function performance for those with ADHD has been found to occur with aging and in females [26]. ASD characteristics may persist across the life span with an increase challenge to social communication [27]. Such challenge is the same across genders, however females show less repetitive and stereotyped behaviour [28]. The independent occurrence of ODD has been found to be invariant across age and gender [29] . The effect of gender and age when these conditions co-occur with FASD is unclear.”

4)The practical relevance in the school context, but also different environments could be illustrated in more detail.

The introduction details how the challenge to psychological processes, in the form of executive function, which is associated with ADHD, ODD, and ASD can impact children within the classroom. Though the focus of this paper is upon an educational context, we agree that it can be relevant to other environments. We have added a paragraph to refer to how the lack of a diagnosis, and consequent lack of understanding of the challenged psychological processes, may permeate across the context of school-exclusion-to-prison-pipeline, impacting on effective support for children with FASD. In a bid to keep the reader focused on the education lens which this study is concerned with, we have kept to a succinct comparison. We refer to this again in the discussion section.

Introduction:

Challenge to psychological processes associated with characteristics of ADHD, ODD and ASD, regardless of diagnosis, may pose an elevated risk of anti-social responses to reward and consequence. The implications of which have been found in the school-exclusion-to-prison-pipeline [8], which indicates that those with undiagnosed neurodevelopmental conditions, including FASD are over-represented in both school exclusion and the youth justice system. A system which is built on reward and sanction as a way of regulating behaviour. Neither education, nor youth justice screen for neurodiversity i.e., the characteristics of a condition without a diagnosis [12]; leading to limited staff training. Knowledge gaps may contribute to lack of support and identification of the ways challenge to psychological processes may impact on behaviour, in both these contexts.”

Discussion:

Challenge to psychological processes in the form of hot executive function may increase the risk of anti-social responses to reward and consequence. These findings may be of interest in the context of the youth justice system, where young people with ADHD, ASD and FASD are over-represented, many of whom may have no diagnosis [8]. The heavy reliance on reward and consequence to shape behaviour within the system, may be ineffective due to the child’s in-ability to comprehend moral and social norms tied to the rules [30], increasing levels of frustration, resulting in anti-social responses. Lack of knowledge regarding the challenge to psychological processes may lead to punitive consequences and further criminalization [12].”

5)The potential for novel interventions could be highlighted more strongly with more concrete examples.

Thank you for this suggestion, and the subsequent one (point 6). This has enabled us to strengthen our rationale for further study into exploring effective educational interventions aimed at increasing school inclusion for children with FASD.

We have addressed point 5 and point 6 (below) by including following into the discussion section:

“Welcome advances in theoretical models that aim to improve support for the most vulnerable school children, such as trauma informed approaches [52] may go some way to addressing the support needs of children with FASD. These approaches acknowledge that psychological processes, such as regulation and inhibition can be challenged as a result of pre-and post-natal exposure to trauma; providing a broader approach that goes beyond a specific diagnostic understanding. Empirical findings on the efficacy of trauma informed models have for children with FASD is limited.

Educational approaches specific to children with FASD are emerging in the UK, with information and training for educators which have been developed in conjunction with those affected by FASD; however there has been limited empirical studies involving practices towards FASD in educational settings [53]. As a result, there has been little effective intervention identified for children with FASD [54]; further evidence-based exploration is required in this field. Co-production models of intervention which harness stakeholder involvement from students and educators have been found to be effective in improving issues such as mental-health, psychological well-being and school inclusion. Psychological theory to underpin the intervention [55]  and peer-involvement [56] has led to successful interventions which benefit cohorts of students in the classroom regardless of diagnostic condition. Taking a heath-based approach to identifying barriers to interventions in educational settings, and ways to overcome them, such as the Behaviour-Change Wheel taxonomy [30] could be beneficial”.

6)Please consider elaborating in more depth on how current conceptual models are advanced. This could help to frame future research development.

Please refer to point 5

7)Title: There are numerous abbreviations in the abstract. Please consider avoiding abbreviations for more clarity.

We have reduced the number of abbreviations in the Abstract and we have amended the title to reflect these comments, with an aim to making the terms accessible to a wider audience. We have also removed the acronyms from the abstract and simplified the terms to make them more accessible to a wide audience. The title is now:

Heated behaviour in the classroom: How do parents of children with FASD perceive the associations between their child’s personality, cognition and rewards experience? A cross-sectional approach to the importance of FASD diagnosis status.”

Reviewer 2 Report

Thank you for giving me the opportunity to review this manuscript.

1) Please descrive the study's design with a commonly used term in the title and the abstract. Was this a cross-sectional study? Fyrthermore, please describe the study design by the PECO (population, exposure, control and the outcome) in the method section.

2) Please describe the setting, locations, and relevant date, includng periods of recruitment, exposure, follow-up, and data collection.

3) Please describe the eligibility criteria, and the sourses and methods of selection of participants. Please describe any matching criteria.

 4) Please clearly describe all potential confounders. Please describe all statistical methods, including those used to control for confounders in thhe statistical analysis section. Please describe unadjusted estimates and, if applcable, confounder-adjusted estimates and their precision. Please make clear which confounders were adjusted for and why they were included.

5) Please descrive any efforts to address potential sources of bias.

6) Please explain how sample size was arrived at. If applicable, please descrive analytical methods taking account of sampling strategy.

7) Please explain how missing data was addressed. Please describe number of participants with missing data for each valuable of interest.

I think it is better to revise the manuscript before publication.

Author Response

Reviewer 2

Thank you very much for your comments, your points have led to a manuscript content which now more clearly identifies for the reader that this is not an experimental study and that we are emphasising associations between the variables rather than constructing causal paths. The emphasis upon the non-experimental or cross-sectional nature of the research is made evident from the Title changes through to the conclusions in in the Discussion.

1) Please descrive the study's design with a commonly used term in the title and the abstract. Was this a cross-sectional study? Fyrthermore, please describe the study design by the PECO (population, exposure, control and the outcome) in the method section (AC, CH).

Thank you for the comments regarding the title. We have amended the title to reflect these comments, with an aim to making the terms accessible to a wider audience. The title is now:

“Heated behaviour in the classroom: How do parents of children with FASD perceive the associations between their child’s personality, cognition and rewards experience? A cross-sectional approach to the importance of FASD diagnosis status.”

Thank you for your comments with regard to using the PECO format. This study was non-experimental and therefore the participants were not exposed to any condition or treatment arm. However, as you suggest we have ensured that the population, control and outcome detail is clear within the Method section. The participants section refers to Population via eligibility criteria, and identifies them as being parents and carers with children age 6-12 years who have suspected or diagnosed FASD. The design section refers to the controls as covariates and identifies them as age and gender. This section also identifies the outcome variable as the helpfulness of rewards in the classroom.

2) Please describe the setting, locations, and relevant date, includng periods of recruitment, exposure, follow-up, and data collection.

Thank you for this. Some of these items were already present in the methods section, in terms of recruitment and data collection, however we have made it clearer and added details to include the period of recruitment and follow up. As the participants were not exposed to any condition this item could not be addressed. The following statements have been added to the Method section.

 “The online survey was made available via FASD networks on social media. The survey was available to participants from April 2022 until Sept 2022.”

 “Where participants emailed expressing interest in the results, they have been advised that a user-friendly version will be made available. Follow up analysis is intended to explore with participants the direction of future research.”

3) Please describe the eligibility criteria, and the sourses and methods of selection of participants. Please describe any matching criteria.

We have added the term eligibility criteria to make it clearer in the methods section referring to participants. As the surveys completed by the participants were used to form the analysis we have added a sentence to make it clear how the surveys were selected. This was also confirmed in the data treatment section of the results. There is no matching criteria in this study.

Method section:

Only surveys where all questionnaires were  >90% completed were eligible to be used within the study.

Results section:

Analysis on the total sample was performed to identify incomplete surveys. Any survey which contained >10% of unanswered questions from a questionnaire was removed from the data collection.”

 4) Please clearly describe all potential confounders. Please describe all statistical methods, including those used to control for confounders in thhe statistical analysis section. Please describe unadjusted estimates and, if applcable, confounder-adjusted estimates and their precision. Please make clear which confounders were adjusted for and why they were included.

Thank you for the suggestion with regard to confounders. We have addressed this in the Method section by making it clearer which observed variables were selected as mediators, moderators and co-variates, and why:

A number of analyses were performed to understand the interactions between the variables and how those relationships were associated with the helpfulness of rewards (see data treatment section). Due to the prevalence of co-occurring ADHD, ODD and ASD the analysis strategy was to use a multi-variate approach to understand how observed variables (executive function, FASD diagnosis, age and gender) may confound  any meaningful interpretation of the relationship between characteristics of each condition and the helpfulness of rewards.

For the main moderated mediation analysis, the predictor variables were scores from the ADHD, ODD or ALT measures. The outcome variable was ratings of the helpfulness of reward systems in schools. Given prior literature indicated there was a relationship between ADHD, ODD, ALT and challenge to executive function the score of executive functions (regulation or inhibition) was used as a mediator to understand its indirect effect in the relationship between the characteristics and rewards. Though literature indicated a relationship between traits and executive function, it was possible that having FASD diagnosis may account for any variance in any indirect effect found, and so the observed variable FASD Diagnosis (suspected or diagnosed) was used as a moderator on the indirect effect. Covariates were age and gender within all of the analyses, due to prior research indicating the effect on ADHD characteristics.”

We have added a section within the data treatment section of Results to further explain how these observed variables were managed within the main analysis:

In order to understand the relationship between ADHD, ODD, ALT and executive function mediation analyses using Hayes (2017) PROCESS Model 4 was considered and discounted due given that this was a non-experimental study which was considering the relationship between the observed variables rather than any perceived causal effect. PROCESS Model 14 was instead employed, allowing the fourth observed variable (FASD Diagnosis) to be added into the analysis. This moderated mediation analyses  considered how EF and FASD diagnosis qualified the relationship between traits and the helpfulness of rewards, whilst controlling for the effect of age and gender. Regulation and Inhibition were individually considered as mediators of these relationships. The final step of these analyses included FASD diagnosis status, confirmed vs suspected, as a putative moderator of the mediation.”

Considerable changes have been made to the results of the moderated mediation, in terms of the Hayes 2017 PROCESS model and the non-causal inference that we took from the results. In doing so, we aim to provide a more specific interpretation of the effect coming from each relationship between variables within model 14. These results provide both a statistical and interpretive explanation of the model, in a bid to reduce any misinterpretation that a causal effect can be inferred from the findings. We have pulled forward from the appendix the reported coefficients for each pathway in the model and included the non-significant effect from age and gender. We have made it clearer that this is a non-causal inference process.

The Discussion content has been added to, ensuring that it is clearer that the results take a non-causal inference:

The significant results from the moderated mediation analyses tell us a story of how the interaction of these measured variables is important for the helpfulness of rewards for children with FASD. However as heeded by Hayes [43], they are not indicative of a causal relationship. As with any non-experimental study, causal inference can lead to an incorrect interpretation of the results. Rather, the conditional processing model allowed consideration of how the variables may relate to one another conditional on a fourth variable. In taking an non-causal inference from the analyses, it recognises that un-observed variables could lead to a misinterpretation of the findings.”

In further ensuring transparency in the interpretation of results, we have added content to the limitation section to include acknowledgement that latent variables are uncontrolled for and therefore could account for variance in the results:

As indicated previously, latent variables uncontrolled for in this non-experimental study could account for some of the variance observed and therefore the results from this study should be inferred as indicative rather than definitive.”

5) Please descrive any efforts to address potential sources of bias.

This valid point was addressed in the survey itself, and we have now made that clear in materials section of the Method, thank you for suggesting this. We have added the following:

To consider biases from attrition and non-completion we have added the following content.

Chi square comparison of the sample of care-givers who completed the survey versus those who did not indicated no difference in the gender proportions, X2 = 3.36, p > .05. However, the proportions with a FASD diagnosis did differ, X2 = 6.403, p = .011. This in-dicated that a greater proportion of care-givers whose child did not have a FASD diagnosis failed to complete the survey. The F test analysis comparing the age of the children whose care-giver completed the survey, versus those who did not complete the survey after re-coding their child’s age did not differ significantly, F (1,56) = 1.354, p > .05.”

Potential response biases related to the presentation of the questionnaire items were also added,

Questionnaire items were presented individually onscreen, thus not only ensuring anonymity but also reducing any acquiescent response biases.”

Questions were positively worded and positively scored in an attempt to reduce the risk of bias in reporting from a potential negative halo effect.”

Selection bias may well have been present as our non-experimental design employed a cross-sectional, existing groups design. We have been clear throughout the content of the implications of the this for the interpretation of the findings.

6) Please explain how sample size was arrived at. If applicable, please descrive analytical methods taking account of sampling strategy

An a priori power analysis was conducted. This has been referred to within the methods section. We have expanded the explanation to include:

“To ascertain the sample size an a priori power analysis was conducted using G*Power. Prior research [26] has typically found a large effect of ADHD and executive control, but with diagnostic levels of ADHD, this study was investigating children with lower levels of ADHD characteristics so a medium effect size was adopted. Leading to a g*Power calculated minimum n = 53.”

7) Please explain how missing data was addressed. Please describe number of participants with missing data for each valuable of interest.

We have made clear in the data treatment section how missing data was managed.

Analysis on the total sample was performed to identify incomplete surveys. Any survey which contained >10% of unanswered questions from a questionnaire was removed from the data collection. In addition, surveys which omitted responses to either age, gender, or FASD diagnosis status was also removed. In the final sample of surveys (n = 121) missing data from each measure of ADHD, ODD, ALT, EF (regulation diagnosed FASD n=2) and helpfulness of reward systems (suspected FASD n=3, diagnosed FASD n=1) was replaced with average scores.”

Round 2

Reviewer 2 Report

I think this manuscript would be suitable for publication.